# *CD47* and *IFT57* Are Colinear Genes That Are Highly Coexpressed in Most Cancers and Exhibit Parallel Cancer-Specific Correlations with Survival

**DOI:** 10.3390/ijms25168956

**Published:** 2024-08-17

**Authors:** Kun Dong, Raghib Nihal, Thomas J. Meyer, Satya P. Singh, Sukhbir Kaur, David D. Roberts

**Affiliations:** 1Laboratory of Pathology, Center for Cancer Research, National Cancer Institute, National Institutes of Health, Bethesda, MD 20892, USA; kun.dong@fda.hhs.gov (K.D.); nihalra@vcu.edu (R.N.); 2CCR Collaborative Bioinformatics, Resource, Office of Science and Technology Resources, National Cancer Institute, National Institutes of Health, Bethesda, MD 20892, USA; thomas.meyer@nih.gov; 3Inflammation Biology Section, Laboratory of Molecular Immunology, National Institute of Allergy and Infectious Diseases, National Institutes of Health, Bethesda, MD 20892, USA; spsingh@niaid.nih.gov

**Keywords:** lung adenocarcinoma, glioma, papillary thyroid carcinoma, primary cilium, CD47, intraflagellar transport-57 (IFT57), RNA sequencing, microsynteny, capping protein inhibiting regulator of actin dynamics (CRACD)

## Abstract

An association between high CD47 expression and poor cancer survival has been attributed to its function on malignant cells to inhibit phagocytic clearance. However, CD47 mRNA expression in some cancers lacks correlation or correlates with improved survival. *IFT57* encodes an essential primary cilium component and is colinear with *CD47* across amniote genomes, suggesting coregulation of these genes. Analysis of The Cancer Genome Atlas datasets identified *IFT57* as a top coexpressed gene with *CD47* among 1156 human cancer cell lines and in most tumor types. The primary cilium also regulates cancer pathogenesis, and correlations between IFT57 mRNA and survival paralleled those for CD47 in thyroid and lung carcinomas, melanoma, and glioma. CD47 ranked first for coexpression with IFT57 mRNA in papillary thyroid carcinomas, and higher expression of both genes correlated with significantly improved overall survival. CD47 and IFT57 mRNAs were coordinately regulated in thyroid carcinoma cell lines. Transcriptome analysis following knockdown of CD47 or IFT57 in thyroid carcinoma cells identified the cytoskeletal regulator CRACD as a specific target of IFT57. CRACD mRNA expression inversely correlated with IFT57 mRNA and with survival in low-grade gliomas, lung adenocarcinomas, and papillary thyroid carcinomas, suggesting that IFT57 rather than CD47 regulates survival in these cancers.

## 1. Introduction

The expression of the cell surface glycoprotein CD47 is elevated in many solid tumors and hematologic malignancies [1,2]. Higher CD47 expression in some cancers correlates with decreased patient survival [3,4,5,6]. The prevailing evidence indicates that high CD47 expression on the surface of cancer cells functions in those cancers as an immune checkpoint by engaging its inhibitory counter-receptor, signal regulatory protein-α, on innate immune cells [2]. Higher expression of CD47 on tumor cells thereby functions as a ‘don’t eat me’ signal that prevents their clearance by macrophages and neutrophils and limits the presentation of tumor antigens to cytotoxic T cells by antigen-presenting cells.

In contrast, higher CD47 expression in cutaneous melanomas is associated with increased overall and disease-free survival and increased tumor infiltration and activation of NK and CD8 T cells [7,8,9]. Our further analysis of The Cancer Genome Atlas (TCGA) RNAseq data has identified several additional cancers, including lung adenocarcinoma and papillary thyroid carcinoma, that exhibit positive correlations between CD47 mRNA expression and patient survival but lack the CD47-associated immune signatures that were identified in melanomas. The mechanism by which elevated CD47 expression leads to improved survival in these cancers is unknown, but it is not consistent with the ‘don’t eat me’ hypothesis, which predicts poorer survival due to decreased innate immune clearance of tumor cells bearing high CD47.

An alternate hypothesis is suggested by an analysis presented here of transcripts coexpressed with CD47 mRNA in TCGA RNAseq data for 29 tumor types, which identified intraflagellar transport-57 (*IFT57*) as a top CD47-coexpressed gene. IFT57 is a component of the primary cilium, which regulates signaling pathways critical for cancer progression [10]. Higher IFT57 mRNA expression in lung adenocarcinomas was associated with improved overall survival [11]. Overexpression of IFT57 sensitized glioblastoma cells to apoptotic cell death [12], whereas silencing IFT57 in H157 lung carcinoma and A549 oral squamous carcinoma cells inhibited cell proliferation and increased apoptosis [13]. Targeted disruption of another primary cilium gene, *ift88,* in the mouse thyroid resulted in spontaneous tumorigenesis [14]. Conversely, knockdown or deletion of *IFT88* in human thyroid cancer cells impaired ciliogenesis and resulted in metabolic reprogramming and mitochondria-dependent apoptosis [15,16]. 

*IFT57* and *CD47* are adjacent colinear genes located on human chromosome 3, and this microsynteny is highly conserved across mammalian, reptilian, and avian genomes (Appendix A). Several signaling pathways have been identified that mediate the increased CD47 expression in specific cancers [17,18,19,20,21,22,23], but the possibility has not been considered that transcription regulators interacting with the promoter region of *CD47* may simultaneously regulate expression of the adjacent *IFT57*. To further examine the degree to which *CD47* and *IFT57* are coregulated in cancers and identify biomarkers to distinguish the functions of IFT57 and CD47 in these cancers, we used CRISPR/Cas9-targeted knockdown of IFT57 or CD47 in a thyroid carcinoma cell line to identify transcripts sensitive to altered *IFT57* expression. 

## 2. Results

### 2.1. Coexpression of IFT57 with CD47 in Human Tumors and Cancer Cell Lines

Examination of CD47 coexpressed genes in 29 TCGA cancer types identified *IFT57* as a gene that is highly coexpressed with CD47 mRNA in carcinomas from multiple tissues (Table 1, Figure 1a–c). *CD47* was the top-ranked gene co-expressed with *IFT57* in breast, bladder, ovarian, endometrial, thyroid, and lung carcinomas. Conversely, *IFT57* was also the top co-expressed gene with *CD47* in several carcinomas. Significant positive correlations between CD47 and IFT57 mRNA expression extended to neuroendocrine tumors, sarcomas, and some hematologic malignancies, but IFT57 and CD47 co-expression had lower Spearman’s coefficients and rank orders in these cancers (Table 1). Apart from rare cancers with documented amplification of the *IFT57* locus, including two ovarian epithelial carcinomas, IFT57 mRNA was expressed at similar levels in all cancers (new Appendix A). Coregulation of *CD47* and *IFT57* is suggested by the proximal colinear location of these genes on human chromosome 3, and by conservation of this microsynteny across diverse amniote genomes (Appendix A). The potential for elements in the *CD47* gene to regulate *IFT57* expression is also suggested by a 2.7-fold increase in IFT57 mRNA compared to WT in NK cells from *cd47^−/−^* mice generated by an insertion that disrupted exon 2 of *cd47* [24] (GSE113980 in [25]).

### 2.2. Coexpression of CD47 and IFT57 mRNAs Is Intrinsic to Cancer Cells

A highly significant positive correlation was also observed between CD47 and IFT57 mRNA expression across the 1156 cell lines in the Cancer Cell Line Encyclopedia (CCLE) [26] (*p* = 4.6 × 10^−36^, Table 1, Figure 1d). This suggests that mechanisms mediating *IFT57* coregulation with *CD47* are intrinsic to cancer cells, but this does not exclude additional contributions from IFT57 co-expression with CD47 mRNA in nonmalignant cells in stroma of the tumors analyzed in TCGA. 

### 2.3. IFT57 and CD47 mRNA Expression in Tumors Generally Have Concordant Correlations with Survival

Multiple publications have reported correlations between higher CD47 mRNA or protein expression in specific cancers and poorer survival [3,4,5,6]. However, none of the TCGA tumor datasets showed significant associations between higher CD47 mRNA expression and decreased overall survival using a mean cutoff (Table 1). Although IFT57 mRNA in lung squamous carcinomas showed coexpression with CD47 (Table 1), a previous study did not find a significant correlation between CD47 protein expression and survival for this cancer [27]. Consistent with that study, CD47 mRNA expression above the mean was not significantly correlated with overall survival in TCGA lung squamous carcinomas, but survival trended lower with higher CD47 mRNA (Table 1, Figure 2a). This trend became significant when the expression cutoff was optimized as described [3] (36 versus 63 months median survival, logrank *p*-value = 5.1 × 10^−3^, Appendix A) or when median expression was used as a cutoff (Appendix A). Low-grade glioma showed a similar decreased survival trend with higher CD47 mRNA that approached significance at a mean cutoff (Table 1, Figure 2c) and became significant with cutoff optimization (Appendix A) or when median expression was used as a cutoff (Appendix A). Notably, IFT57 mRNA higher than the mean was significantly correlated with decreased overall survival in low-grade glioma (55 versus 93 months, logrank *p*-value = 2.7 × 10^−4^) and in lung squamous carcinoma (39 versus 64 months, logrank *p*-value = 9.9 × 10^−3^ Table 1, Figure 2b,d). IFT57 mRNA higher than the median was also significantly correlated with decreased overall survival in low-grade glioma and lung squamous carcinoma (Appendix A). Head and neck squamous cell carcinoma also showed significant correlations between IFT57 mRNA expression higher than the mean and decreased overall survival that was not significant for CD47 (Appendix A).

Conversely, CD47 mRNA expression above the mean in lung adenocarcinoma significantly correlated with improved overall survival (53 versus 41 months median survival, *p* = 0.021, Figure 2e), but IFT57 above the mean was more strongly associated with improved survival (58 versus 42 months median survival, *p* = 4.6 × 10^−3^, Figure 2f). This was consistent with IFT57 being the top-ranked gene for coexpression with CD47 in lung adenocarcinoma (Table 1) and a previous analysis of TCGA data [11]. Neither CD47 nor IFT57 mRNA expression was significantly correlated with overall survival in lung adenocarcinoma using a median cutoff (Appendix A). Breast carcinomas and uveal melanoma also showed significant correlations between higher IFT57 expression above the mean and improved overall survival that were not significant for CD47 (Appendix A). Taken together, these data indicate that IFT57 mRNA generally exhibits stronger correlations with improved or decreased cancer survival than CD47 mRNA expression. This suggests that, in some tumor types, IFT57 rather than the coregulated CD47 mRNA expression could be the dominant driver of survival.

Cutaneous melanomas exhibit a strong positive correlation between CD47 mRNA expression and improved survival, which relates to enhanced NK and CD8 T cell activation in tumors with elevated CD47 [7,8,9]. IFT57 mRNA expression above the mean was also significantly correlated with improved overall survival, but with a higher *p*-value than for CD47 (Table 1). Correspondingly, CD47 was ranked 432nd for coexpression with IFT57 in cutaneous melanoma (Table 1). 

CD47 was top-ranked for IFT57 mRNA coexpression in papillary thyroid carcinomas, with a Spearman coefficient r = 0.62 (Table 1). Papillary thyroid carcinomas exhibited strong associations between survival and expression of CD47 or IFT57. IFT57 mRNA expression greater than the mean showed significant increases in both overall and disease-free survival compared to tumors with IFT57 expression lower than the mean (*p* = 2.0 × 10^−3^ and 0.019, respectively), and CD47 mRNA expression above the mean was associated with improved overall survival CD47 (logrank *p*-value = 3.8 × 10^−5^, Figure 3a,c,d). Disease-free survival trended longer with higher CD47 but did not achieve significance (*p* = 0.28, Figure 3b). Current TCGA data for other types of thyroid cancer are limited. Because existing TCGA data for anaplastic thyroid tumors do not include RNAseq data, correlations with survival could not be examined. Papillary thyroid carcinomas did not show significant CD47 mRNA correlations for the immune markers identified in cutaneous melanomas.

### 2.4. Coregulation of IFT57 and CD47 in Thyroid Carcinoma Cells

The bromodomain extra-terminal inhibitor JQ1 suppressed the elevated CD47 expression in double-hit B cell lymphoma cells [22]. Similarly, treatment of MDA-T68 thyroid carcinoma cells with 10 μM JQ1 significantly decreased mRNA for long and short isoforms of CD47, with a parallel time-dependent decreasing trend in mRNA expression for IFT57 mRNA (Figure 4a–c). Conversely, the BRAF^V600E^ inhibitor vemurafenib increased CD47 expression in melanoma cells [23]. Treatment of MDA-T68 cells with 10 μM vemurafenib resulted in significant increases in CD47 and IFT57 mRNA (Figure 4d). Thus, known positive and negative transcriptional regulators of CD47 mRNA expression in cancer cells have parallel effects on IFT57 mRNA expression.

The CCLE contains 11 carcinoma cell lines of thyroid origin, but only one is a papillary thyroid carcinoma [26]. Four were derived from tumors with follicular and six with anaplastic histology. IFT57 and CD47 mRNA expression were positively correlated across the 11 thyroid carcinomas but did not achieve significance (Appendix A, R^2^ = 0.18, *p* = 0.2). However, a significant correlation was found for the 5 primary anaplastic thyroid carcinoma cell lines: SW579, 8305C, MB1, CAL62, and 8505C (R^2^ = 0.84, *p* = 0.028, Figure 4e). CD47 and IFT57 mRNA expression were consistently higher in 8505C anaplastic thyroid carcinoma cells relative to Nthy-ori-3-1 normal thyroid follicular epithelial cells and other available thyroid carcinoma cell lines (Figure 4f–h). Based on these data, we selected 8505C cells to identify genes that are differentially regulated by CD47 or IFT57 in thyroid cancers.

### 2.5. Differential Knockdown of IFT57 and CD47 in Thyroid Carcinoma Cells

Preliminary experiments using transient siRNA knockdown of IFT57 showed no significant effects on CD47 mRNA expression in 8505C cells (Appendix A). Transient knockdown of IFT57 mRNA also did not significantly alter CD47 protein expression (Appendix A). Conversely, transient overexpression of IFT57 by plasmid transfection resulted in a lower increase in CD47 mRNA expression in MDA-T68 follicular variant of papillary thyroid carcinoma cells than induced by transfection with the empty vector (Appendix A). These results exclude significant cross regulation of *CD47* gene expression by IFT57-dependent primary cilium signaling.

Stable CRISPR/Cas9 mutants were prepared by targeting *IFT57* or *CD47* in 8505C cells. We were unable to isolate clones that completely lacked IFT57 expression, but two clones were selected that had consistent >60% knockdown of IFT57 mRNA and <20% decreased CD47 mRNA (Figure 5a,b). Reduced IFT57 protein expression was confirmed by Western blot, whereas cell surface CD47 protein expression assessed by flow cytometry was unchanged (Figure 5d,e). Using CD47 guide RNAs, an early passage pool of cells was obtained in which IFT57 mRNA expression was unchanged and cell surface CD47 expression was undetectable by flow cytometry (Figure 5c,d). PCNA mRNA expression was reduced approximately two-fold in both IFT57 knockdown clones (Figure 5f). Correspondingly, MTS assays indicated moderately decreased proliferation rates for the two IFT57 knockdown clones compared to the parental cell line (Figure 5g). 

### 2.6. Identification of IFT57-Dependent Gene Expression in Thyroid Cancers

Bulk RNA sequencing was used to identify genes that are sensitive to altered CD47 or IFT57 expression in 8505C cells. Because the initially obtained CD47 knockout clones tended to regain CD47 expression with repeated passage, an early passage pool that was negative for cell surface CD47 was used to prepare a library. Two independent IFT57 knockout clones that exhibited the most consistent phenotypes were used for this analysis. 

Consistent with the divergent functions of CD47 and IFT57 in other cell types, there was limited overlap between genes with significantly altered mRNA expression in the CD47 and IFT57 knockdowns (Figure 6a). Using a *p* < 0.05 cutoff, 1170 genes had altered mRNA expression in both of the IFT57 knockdown clones, and 1129 of these lacked significant changes in the CD47 knockdown pool cells. Of the 41 overlapping genes (Figure 6a, Appendix A), 11 were significantly correlated with mRNA expression of both CD47 and IFT57 in TCGA papillary thyroid carcinomas, but only 5 of the latter were consistent with the direction of change in the IFT57 knockdown cell lines (Appendix A). As expected, CD47 was one of the 92 mRNAs altered only in the CD47 knockdown pool and was significantly decreased (adjusted *p* = 9.8 × 10^−8^). Of the 67 annotated genes with available TCGA papillary thyroid carcinoma data, 46 had significant coexpression with CD47 in the tumors. Of these, only 6 had equal or lower *p*-values for coexpression with IFT57 in the tumors, and 27 had significant coexpression only with CD47 (Appendix A). This confirmed that knockdown of CD47 had limited effects on the expression of most IFT57 target genes. 

We were more interested in identifying genes with expression that is selectively regulated by IFT57 in thyroid cancer cells. Pathway analysis comparing transcriptome data for the IFT57 knockdown clones with WT 8505C cells identified significant alterations in mRNAs involved in cell motility, differentiation, and extracellular matrix regulation including fibronectin (FN1) (Appendix A, Appendix A). Elevated fibronectin mRNA was confirmed in six independent IFT57 knockdown clones by qPCR (Appendix A). Increased fibronectin protein expression was confirmed by Western blot for the clones used for RNA sequencing (Appendix A). Although 1129 of the IFT57-dependent genes were unchanged in the CD47 knockdown cells, some CD47-dependent genes may have been missed due to incomplete knockdown. Comparison with a published dataset of 4612 CD47-dependent genes identified using WT and *cd47^−/−^* mouse CD8 T cells (GSE239430) identified limited overlap between CD47-dependent genes in mouse CD8 cells and IFT57-dependent genes in human 8505C cells (Appendix A). 

To identify IFT57-dependent genes in 8505C cells that could be used as biomarkers and may also be relevant to the survival effects of IFT57 expression in papillary thyroid carcinoma, the IFT57-dependent genes identified in 8505C anaplastic thyroid carcinoma cells were screened for coexpression with IFT57 and CD47 in the TCGA papillary thyroid carcinoma tumor data. From the 1129 IFT57-dependent genes, we identified 202 genes that differed at least 8-fold compared to WT, with 73 up- and 129 down-regulated in the two IFT57 knockout clones. Of these, 34 showed positive or negative coexpression with IFT57 mRNA expression in TCGA papillary thyroid tumors with a *p*-value < 10^−5^ (Table 2). Although expression of these genes was sensitive to loss of IFT57 but not CD47 in thyroid carcinoma cells, a majority showed positive or negative coexpression with CD47 mRNA in the tumors that paralleled their IFT57 coexpression (e.g., RAMP1, Figure 6b). This is consistent with the hypothesized coregulation of *IFT57* and *CD47* transcription in the tumors by cis-acting factors and identified *RAMP1* as one of the 174 CD47/IFT57 codependent genes in Appendix A. Presumably, coexpression of these genes in thyroid tumors is mediated by factors that coordinately induce both CD47 and IFT57 mRNA expression. 

To identify specific targets of IFT57, we focused on genes in Figure 6b that showed minimal coexpression with CD47 in thyroid tumors. Two of the genes that were up-regulated in IFT57 knockdown cells, *CRACD* (Capping protein inhibiting Regulator of ACtin Dynamics) and *GAS6*, showed stronger negative correlations with IFT57 than with CD47 expression in the tumors. CRACD mRNA expression lacked significant correlation with that of CD47 (*p* = 0.95), identifying it as the best candidates for a specific biomarker of IFT57 function (Figure 6b). Conversely, several of the genes that were down-regulated in the IFT57 knockdown cells including *NMB* exhibited better positive correlations with IFT57 relative to CD47 mRNA in tumors and were additional candidates for IFT57-dependent markers (Figure 6b). Notably, these candidate IFT57-dependent genes have reported functions in other cancers, and roles for GAS6 [28,29] and NID2 have been reported in thyroid cancer [30].

Real-time qPCR was used to confirm the sensitivity of the selected transcripts to IFT57 or CD47 knockdown in 8505C cells. As expected, RNF180, NID2, and NMB mRNA expression were decreased in the two IFT57 knockdown clones (Figure 6c–e). In contrast, the expression of RNF180 was unchanged, and NID2 and NMB mRNAs were elevated in the CD47 knockdown cells, indicating that these are specific positively regulated targets of IFT57. Conversely, strong up-regulation of CRACD mRNA in the two IFT57 knockdown clones but not in the CD47 knockdown cells validated this gene as a specific target of inhibitory IFT57 function (Figure 6f). GAS6 mRNA also showed the expected significant up-regulation in the IFT57 knockdown clones but not in the CD47 knockdown cells, indicating that GAS6 is another IFT57-specific target (Figure 6g).

Consistent with the mRNA expression, intracellular CRACD protein expression assessed by flow cytometry was elevated relative to that in WT 8505C cells in the two IFT57 mutants but lower in the CD47 mutant pool (Figure 6h). We tested the effects of two CD47 ligands to further confirm that CRACD is an IFT57-specific target. Thrombospondin-1 treatment for 4 h decreased basal CRACD mRNA expression in the WT cells and the CD47 and IFT57 mutants, suggesting that this response is mediated by a thrombospondin-1 receptor other than CD47 (Appendix A). The antibody B6H12 is known to block CD47 interactions with thrombospondin-1 and its counter-receptor SIRPα. Treatment with this antibody had no effect on CRACD expression in WT cells (Appendix A).

### 2.7. Correlations of IFT57 Target Gene Expression with Papillary Thyroid Carcinoma Survival

Consistent with its inverse coexpression with IFT57 in thyroid carcinoma cells in vitro (Figure 6b,f), CRACD mRNA expression above the mean was associated with significant decreases in both disease-free and overall survival in TCGA papillary thyroid carcinomas (Figure 7a,b). Consistent with their positive coexpression with IFT57 in vitro, NMB or RNF180 mRNA expression greater than the mean achieved significant logrank *p*-values versus expression less than the mean for increased disease-free but not overall survival (Appendix A). 

### 2.8. Correlation of CRACD mRNA Expression with Survival in TCGA Cancers

To evaluate the broader utility of CRACD as an IFT57-dependent biomarker in tumors, associations between CRACD expression and overall survival were examined in the 29 TCGA tumor datasets (Table 1). CRACD expression greater than the mean achieved significance in 6 of the 29 tumor types, including 3 where IFT57 was most significantly associated with overall survival: lower-grade glioma, lung adenocarcinoma, and papillary thyroid carcinoma. Consistent with the inhibition of CRACD mRNA by IFT57 in cells, IFT57 and CRACD mRNA had inverse correlations with survival in each of these tumor types. Lung adenocarcinoma also showed an association of higher CRACD mRNA with decreased disease-free survival (21 versus 41 months, *p* = 0.023, Figure 7c). Consistent with the improved overall survival associated with higher IFT57, higher CRACD mRNA in low-grade glioma was associated with decreased overall survival (67 versus 114 months, *p* = 4 × 10^−4^, Figure 7d). Two renal carcinoma datasets and a pediatric AML dataset showed significant correlations between CRACD and overall survival that may be independent of IFT57 regulation (Table 1).

## 3. Discussion

Studies of genomic evolution have identified numerous examples of cis-acting regulatory mechanisms that coordinate the expression of adjacent genes [31,32]. The highly conserved microsynteny for *IFT57* and *CD47* across amniote genomes is consistent with conserved coordinated regulation of their expression. The physiological selection pressures that have maintained this microsynteny remain unknown, but recognition of this coregulation is important for interpreting many recent publications that have identified transcriptional regulators relevant to cancer that control the expression of CD47 and correlate with cancer survival or responses to therapy [17,18,19,20,21,22,23]. The present results confirm that at least two known pharmacologic regulators of CD47 mRNA expression in other cancers [22,23] similarly regulate the expression of IFT57 mRNA. The high degree of coexpression for these genes across multiple tumor types and in the 1156 cancer cell lines in CCLE further supports their cell-autonomous coordinated regulation in cancer cells. Several of the identified selective targets of IFT57 showed significant correlations between expression and patient survival. Because IFT57 showed stronger correlations than CD47 with poorer survival in several cancers, the assumption that the ‘don’t eat me’ function of CD47 accounts for such correlations may need to be reevaluated.

We identified several genes that are specifically regulated by IFT57 but not CD47 knockdown in the 8505C thyroid carcinoma cells, and of these, *CRACD* showed the highest specificity for dependence on IFT57 but not CD47 mRNA expression in papillary thyroid tumors. Thus, CRACD may be a specific biomarker for IFT57 function in the primary cilium that is relatively insensitive to CD47 expression. In contrast, most of the genes identified to be dependent for their mRNA expression only on IFT57 in thyroid carcinoma cells were nonetheless significantly coexpressed with CD47 and IFT57 in papillary thyroid tumors. This divergence could involve upstream factors that modulate IFT57 target genes secondary to coregulating CD47 and IFT57 expression either in the thyroid carcinoma cells or in tumor stromal cells (Figure 1). The mechanisms by which IFT57 regulates the expression of these genes remains to be determined. Whether any of the identified IFT57-dependent genes contribute to the observed improved survival remains unclear.

The present results identify distinct gene sets that are regulated by decreasing the expression of CD47 or IFT57 in thyroid carcinoma cells in vitro, but many of these genes exhibit strong correlations with both CD47 and IFT57 mRNA expression in thyroid tumors. This coregulation may result from the proximity of *CD47* and *IFT57* genes and the presence of super enhancers proximal to *CD47* [18]. The strong coexpression of CD47 and IFT57 mRNAs across carcinomas from multiple tissues suggests that engagement of these or other nearby enhancers by oncogenic transcription factors induces the expression of both genes (Figure 1). 

The coregulation of IFT57 with CD47 defined here in thyroid carcinomas generalizes to survival effects in some additional cancer types but is not universal to all malignancies. This indicates the need to further evaluate the molecular basis for the many reported correlations between CD47 expression and cancer prognosis. The antiphagocytic function of CD47 contributes to the association of poor survival with higher CD47 expression in some cancers [3,4,5,6], but we cannot exclude that coregulation of IFT57 also contributes to the survival deficits in tumors with high CD47 expression. Based on evidence for primary cilium functions in multiple cancers and the correlations between IFT57 expression and survival in several cancers [10,11,12,13], targeting the IFT57 and CD47 axes together may enhance the efficacy of the CD47-specific therapeutics currently in clinical trials. 

In addition to serving as a biomarker that is dependent on IFT57 expression, further studies are required to determine whether CRACD contributes to the observed survival outcomes. CRACD has documented effects in other cancers, mediated by the cytoskeleton and β-catenin pathway, that could account for the altered migration and invasion of the IFT57 knockdown clones [33]. However, recent unreviewed work suggests additional effects of CRACD on pathways relevant to cancer including differentiation, stem cell maintenance and immune evasion [34,35]. Additional studies are needed to determine whether CRACD and other targets of IFT57 identified here contribute to the correlations between CD47 mRNA expression and survival in various cancers.

Studies are also needed to determine the extent to which CD47 and IFT57 are coregulated in nonmalignant cells in the tumor microenvironment. The differential expression of more than 10% of the top 202 IFT57-specific target genes identified in 8505C cells in WT versus *cd47^−/−^* naïve murine CD8 T cells (GSE239430) suggests that such coregulation occurs in immune cells. These include RAMP1 mRNA, which was positively correlated with CD47 and IFT57 mRNA expression in the tumors. Others have identified RAMP1 expression as a prognostic factor that is related to immune infiltration for some cancers [36]. Components of the intraflagellar transport complex including IFT57 are critical for trafficking of the T cell receptor to the immune synapse [37,38]. 

These data necessitate reexamination of the oncogenic signaling pathways that have been reported to modulate cancer pathogenesis and responses to therapies that alter CD47 expression [17,18,19,20,21,22,23]. Does coregulation of IFT57 also play a significant role in the observed responses or result in a therapeutic response that is independent of CD47? The interpretation of positive or negative correlations between CD47 expression and cancer survival should also be reevaluated. We focused here on thyroid carcinomas where higher CD47 expression is associated with improved outcomes, but the stronger associations of higher IFT57 mRNA with poorer survival in lung squamous carcinoma, glioma, and head and neck squamous carcinoma raise concern that the previous studies attributing impaired survival of cancers with higher CD47 expression exclusively to its ‘don’t eat me’ function should also consider the role of the coexpressed IFT57 in patient survival and responses to therapy [3,4,5,6].

## 4. Materials and Methods

### 4.1. Reagents and Cell Lines

The follicular variant of papillary thyroid carcinoma cell line MDA-T68 and the papillary thyroid carcinoma cell lines MDA-T41 and MDA-T85 [39] were purchased from ATCC (Manassas, VA, USA). The 8505C undifferentiated anaplastic thyroid carcinoma cell line and Nthy-ori 3-1 normal human thyroid follicular epithelial cells were purchased from Sigma-Aldrich (St. Louis, MO, USA). Human platelet thrombospondin-1 was prepared as described [40]. The function-blocking CD47 antibody B6H12 (Functional Grade, eBioscience, San Diego, CA, USA) was obtained from ThermoFisher Scientific (Waltham, MA, USA). IFT57 antibody for flow cytometry and Western blot (MA525044) was purchased from ThermoFisher Scientific. Fibronectin polyclonal antibody (FN1) was purchased from Proteintech (Rosemont, IL, USA) (15613-1-AP). Polyclonal rabbit antibodies against CRACD (PA5-61669 and PA5-60292) were purchased from Life Technologies|AB/Invitrogen (Waltham, MA, USA). β-Actin antibody was purchased from Sigma (A2228). Anti-α-tubulin was purchased from Cell Signaling Technology (Danvers, MA, USA) (3873). Human IFT57 expression plasmid was purchased from GenScript (Piscataway, NJ, USA) (OHU28914D). Lipofectamine 2000 reagent was purchased from Life Technologies Corporation (11668027). The bromodomain inhibitor JQ1 was purchased from APExBIO (Houston, TX, USA) (A1910). Vemurafenib (PLX4032) was purchased from Selleck Chemicals (Houston, TX, USA) (S1267). CellTiter 96 Aqueous One Solution Cell Proliferation Assay (MTS) was purchased from Promega (Madison, WI, USA) (G3580).

### 4.2. Generation of IFT57KO and CD47KO Cell Lines via CRISPR 

The Cas9-GFP plasmid pSpCas9(BB)-2A-GFP was purchased from Addgene (Watertown, MA, USA) (#48138). Three gRNAs targeting human IFT57 sequences 5′-CGTCGTCACGACGTCGGGTT-3′ (sgRNA-1), 5′-CAAACCCGACGTCGTGACGA-3′ (sgRNA-2), and 5′-TACAGCCACGACCGGTTACC-3′ (sgRNA-4) were designed (Appendix A). The guide RNA sequences were cloned into Cas9-GFP plasmid (Addgene) to obtain three respective IFT57 targeting sgRNA-Cas9-GFP clones. 8505C cells in a 6-well plate were transfected with each gRNA plasmid. In addition, 8505C cells were co-transfected with both sg1 and sg4-Cas9-GFP plasmids using lipofectamine 2000 (Life Technologies Corporation) [41]. The cells were sorted by FACS to isolate the top ~3% of GFP positive cells, which were plated individually into 96-well plates using the sorter. When the clones sufficiently proliferated, a second round of sorting was performed based on IFT57 expression to identify IFT57-low single-cell clones and confirmed by Western blot.

Disruption of CD47 was performed using a human CD47 gRNA targeting sequences 5′-CGACCGCCGCCG CGCGTCACAGG (intron) and 5′CAGCAACAGCGCCGCTACCAGGG (first exon) [42]. 8505C cells were co-transfected with two gRNA Cas9-GFP plasmids [41] using lipofectamine 2000 according to the manufacturer’s instructions. The cells were sorted by FACS to isolate the top ~3% of GFP-positive cells and plated into 96-well using the sorter. When the cells had sufficiently proliferated, a second round of sorting was performed based on CD47 expression to isolate CD47-low and CD47-null populations using CD47-PE antibody (Biolegend, San Diego, CA, USA). The cells were expanded, and loss of CD47 expression was re-validated by flow cytometry analysis using anti-CD47-APC (Biolegend).

Wild-type (WT) 8505C cells, six clones of IFT57 depleted cells (sg1-F4, sg2-B6, sg2-C8, sg2-H1, sg4-H11, sg(1+4)-G10), and pooled CD47 knockdown (KO) cells were selected for the current study. WT and mutant 8505C cell lines were cultured in complete RPMI 1640 medium supplemented with glutamine, 10% fetal bovine serum (FBS), and penicillin/streptomycin (ThermoFisher Scientific). Cells were passed using Trypsin EDTA (Gibco, Life Technologies, Waltham, MA, USA) for subculturing. Cells were used at ~80–90% confluency for all RT-PCR experiments.

### 4.3. Bulk RNA Sequencing and Data Analysis

WT 8505C cells, sg1-F4 and sg2-B6 clones of IFT57 depleted cells, and pooled CD47 KO cells were used for bulk RNA sequencing. The WT, IFT57KO, and CD47KO cell lines were cultured using complete RPMI medium for 24 h. The cell pellets (*n* = 4) for each sample were harvested and frozen at −140 °C. Total RNA extraction and mRNA sequencing using mRNA library preparation (poly A enrichment) was performed by Novogene, Sacramento, CA, USA. RNA sequencing data analysis was performed using the CCBR-NIDAP platform as described [7]. Data are available on the Gene Expression Omnibus (GEO) database (GEO Accession: GSE254982). 

Differentially expressed genes in common between WT versus IFT57KO and WT versus CD47KO were curated, and Venn diagrams were created using FunRich 3.1.3 software. Genes in the IFT57KO DEG list (Appendix A) were analyzed using ShinyGO v0.741 [43] and Qiagen Ingenuity Pathway Analysis.

### 4.4. Real-Time qPCR Analysis

Approximately 500,000 cells/mL were plated overnight using 6-well plates. The cells were washed with PBS, and total RNA was extracted using the TRIzol method [44]. A measure of 1 μg of RNA was used to prepare cDNA using the Maxima First Strand cDNA Synthesis Kit (ThermoFisher) according to the manufacturer’s instructions.

Real-time PCR was performed using FastStart Universal SYBR Green (Roche, Basel, Switzerland) as described previously [45] using primers for 18S rRNA, KIAA1211/CRACD, RNF180, NMB, GAS6, and NID2 (Appendix A, Integrated DNA Technologies, Coralville, IA, USA). All treatments were analyzed for mRNA expression of the above genes using technical replicates (*n* = 3) and reproduced in at least two independent experiments. The fold change was calculated using 18S ribosomal RNA as a control with BioRad CFX 96 Real-Time PCR software version 1.0. *p*-values were calculated using the default setting of Bio-Rad CFX Mastro or using ANOVA with replicates.

### 4.5. Cell Proliferation Assay

8505C cells were plated into a 96-well plate with 5000 cells per well in culture medium. The relative numbers of viable cells were determined after 0, 24, 48, 72 and 96 h, respectively, using the Promega CellTiter 96 Aqueous One Solution Cell Proliferation Assay (MTS, Madison, WI, USA) according to the manufacturer’s protocol. 

### 4.6. CD47 Engagement and Blockade Studies

WT, sg1-F4, sg2-B6, and CD47KO cells (~500,000/mL) were plated in 6-well plates with complete RPMI medium and incubated overnight. The next day, the cells were treated with thrombospondin-1 protein (1 μg/mL) for 4 h, using untreated cells as controls for each treatment. Total RNA was extracted, and mRNA expression of CRACD was analyzed via real-time qPCR as described above. 

### 4.7. Flow Cytometry Analysis 

WT and mutant cells were cultured as described above. For staining, cells were suspended in 100 μL of FACS buffer (Hanks Balanced Salt Solution, Mediatech, Waltham, MA, USA) containing 4% FBS (GeminiBio, West Sacramento, CA, USA) and incubated with FITC anti-human CD47 or isotype control antibody (BioLegend) for 30 min at room temperature. Cells were washed twice with FACS buffer and resuspended in 350 µL of FACS buffer for analysis. For intracellular staining, cells were fixed and permeabilized using the Foxp3 staining buffer set (eBioscience) and stained with rabbit anti-CRACD antibody (Thermo Fisher Scientific) followed with Alexafluor 568 goat anti-rabbit IgG antibody according to the manufacturer’s instructions. All flow cytometry was performed using Fortessa flow cytometers (BD Biosciences, Franklin Lakes, NJ, USA), and the data were subsequently analyzed using FlowJO software version 10.10.0 (TreeStar, Woodburn, OR, USA).

### 4.8. TCGA Data Analysis

TCGA datasets with sufficient RNAseq data normalized to diploid cells for papillary thyroid carcinoma and other cancers and for the Cancer Cell Line Encyclopedia [26] were analyzed using cBioPortal tools to determine correlations between IFT57 and CD47 mRNA expression and survival [46,47]. Unless otherwise specified, mean expression was used to distinguish tumors with high versus low expression for survival analyses. Survival analysis using TCGA and GTEx data was also performed using the GEPIA2 (Gene Expression Profiling Interactive Analysis) web server [48], with median expression used to distinguish tumors with high versus low expression. 

### 4.9. Western Blots

Approximately 500,000 cells/mL were plated overnight using 6-well plates. The cells were washed with cold PBS followed by adding RIPA lysis buffer (Thermos Scientific, 89900) containing proteinase and phosphatase inhibitors (Thermo Scientific, 78442). The cells were scraped, and cell lysates were centrifuged at 17,950× *g* for 10 min at 4 °C. The supernatants were collected into new Eppendorf tubes, and 5 µL samples were taken for protein measurement (Pierce™ BCA Protein Assay Kits). Lysate samples containing loading buffer and β-mercaptoethanol were heated at 95 °C for 5 min. SDS-PAGE was performed using Bis–tris 10% acrylamide gels with MES running buffer (Life Technologies). Primary antibodies against IFT57 (1:500), FN1 (1:1000), α-Tubulin (1:2000), β-actin (1:5000) and secondary horseradish peroxidase-conjugated anti-mouse/Rabit IgG antibodies 1:5000 were used for Western blots.

## Data Availability

The data presented in this study are available in the provided Appendix A, in the NCBI’s Gene Expression Omnibus accessible through GEO Series accession GSE254982, and TCGA datasets accessible using tools at https://www.cbioportal.org/ accessed on 3 June 2024. Datasets, code, and other resources needed to reproduce some results in this manuscript are available at https://github.com/NIDAP-Community/Knockout-of-CD47-and-IFT57-in-thyroid-cancer.

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
