# Peer review of "CD47 and IFT57 Are Colinear Genes That Are Highly Coexpressed in Most Cancers and Exhibit Parallel Cancer-Specific Correlations with Survival"

_ijms, 2024, doi:10.3390/ijms25168956_

Round 1
Reviewer 1 Report
Comments and Suggestions for Authors
The manuscript investigates the reasons for the alternating regulation of tumor survival by CD47 in various cancers. ITF57 was identified as an important factor co-expressed with CD47, together with ITF57 target genes. The manuscript is interesting and the study design is accurate. However, the following points should be addressed in a revision of the manuscript:
Line 142: Please correct ´´p = .021´´.
Line 183: Please replace ´´the mutant BRAF inhibitor vemurafenib´´ by ´´the BRAFV600E inhibitor vemurafenib´´.
Line 219: Please correct ´´mutants was prepared´´.
GAS6 binds RTKs and was identified as an IFT57-specific target. Please discuss possible correlations of the ITF57 target GAS6 with vemurafenib activity (upregulation of CD47 and ITF57).
Comments on the Quality of English Languagen..a.
Author Response
Comments and Suggestions for Authors
The manuscript investigates the reasons for the alternating regulation of tumor survival by CD47 in various cancers. ITF57 was identified as an important factor co-expressed with CD47, together with ITF57 target genes. The manuscript is interesting and the study design is accurate. However, the following points should be addressed in a revision of the manuscript:
Line 142: Please correct ´´p = .021´´.
Response 1: corrected
Line 183: Please replace ´´the mutant BRAF inhibitor vemurafenib´´ by ´´the BRAFV600E inhibitor vemurafenib´´.
Response 2: corrected
Line 219: Please correct ´´mutants was prepared´´.
Response 3: corrected
GAS6 binds RTKs and was identified as an IFT57-specific target. Please discuss possible correlations of the ITF57 target GAS6 with vemurafenib activity (upregulation of CD47 and ITF57).
Response 4: The reviewer raises an interesting hypothesis regarding a possible function of the IFT57 target GAS6. This relates to how changing IFT57 expression could broadly perturb gene expression by altering RTK signaling in cells. However, the relevance of this hypothesis to our verafinib data is unclear because our intent was to ask whether a pathway inhibitor known to alter CD47 expression in other cell lines would similarly perturb IFT57 expression, not the downstream signaling controlled by IFT57. Vemurafenib was tested in thyroid lines that were BRAFV600E positive and negative, We were initially surprised that the response shown in the figure was stronger in a V600E negative line. That indicates that the changes in CD47 and IFT57 mRNA expression are BRAF-independent. We are not surprised that vemurafenib has off target effects, and our point is merely that a drug known to alter CD47 expression also alters IFT57 expression.
Reviewer 2 Report
Comments and Suggestions for Authors
The manuscript is built up on an interesting initial observation, the striking co-expression of CD47 and IFT57, both on chromosome 3 p13.12 - only separated by a non-coding RNA gene. The authors show the genes are co-expressed, yet remain independent regulatory response upon various changes and treatments.
There are only a few unclear issues in the manuscript that should be stelled:
How do you explain that knocking down CD47 or IFT57 results in a rather significant number of differentially expressed genes in mRNA sequencing? NOne are transcriptional regulators, co-activators or repressors, nor epigenetic readers writers, or erasers... this may need a bit more elaboration. Thats particularly interesting to explore further as some of the differential genes are expressed by quite significant old changes (like CRECD upon KD of CD47; this is truly amazing but also mysterious).
How come that altered expression of IFT57 results in changes "in cell motility, differentiation, and extracellular matrix regulation including fibronectin" (line 283)." What are the likely regulatory mechanisms affected? Are there any hypotheses? How would the IFT57-"response" genes look if another cell line would be used for analysis? And would there be an overlap to other cell- or tumor types? Is expression of FT57 specific to epithelial cancers (carcinoma) ?
When checking CRACD expression (= one of the IFT57 "target" genes) in the TCGA data base, by using cBioPortal browser, it does NOT pop up as anti-correlated with IFT57 or CD47 in any of the major tumor types, as you would possibly expect for a "IFT57-downregulated target". It appears only slightly anticorrelated with IFT57 (Spearman -0,21).
Furthermore, how is CRACD expression in cancers, especially in thyroid carcinoma, regulated? Is there anything known about the major potential drivers of its expression (or repression)? It is okay that the authors do not speculate wirldly on how CRACD cold be possibly regulated by a gene that isnt a transcriptional regulator, but this leaves the reader somehow unsatisfied. Which alternatives exist by which a structural gene (IFT57) could have an impact on gene expression (not function) of other genes? Maybe this could be more elaborated. (I do not have a solution, unfortunately).
smaller things:
the "dont eat me" hypothesis concerning CD47 needs to be elaborated in the text (lines 54/55). (Like this one, from Wikipedia:" CD-47 acts as a don't eat me signal to macrophages of the immune system which has made it a potential therapeutic target in some cancers, and more recently, for the treatment of pulmonary fibrosis". )
The correlations between CD47 and IFT57 shown in Table 1 should be in some logical order: why not sort by Spearman/Pearson correlation - highest first?
Correlation and Survival data for these genes and tumour types should also be confirmed independently, using another database. Ideally, one that doesnt use exactly the same TCGA data.
In addition, it may be interesting to further explore the survival data. The GEPIA2 browser (GEPIA 2 (cancer-pku.cn), although also using TCGA data, provides very convincing survival plots, and can distinguish readily between overall and disease- or event-free survival, different margins (qartiles, manually selected group assignments), etc. This could show if the survival data shown in Fig. 2 are confirmed. When checking this, I found different survival values compared to those shown in the Fig. 2.
Please explain why you think you were not able to get stronger knock-down CD47 and IFT57 when using CRISPR/cas9 ?
Author Response
Comments and Suggestions for Authors
The manuscript is built up on an interesting initial observation, the striking co-expression of CD47 and IFT57, both on chromosome 3 p13.12 - only separated by a non-coding RNA gene. The authors show the genes are co-expressed, yet remain independent regulatory response upon various changes and treatments.
There are only a few unclear issues in the manuscript that should be stelled:
How do you explain that knocking down CD47 or IFT57 results in a rather significant number of differentially expressed genes in mRNA sequencing? NOne are transcriptional regulators, co-activators or repressors, nor epigenetic readers writers, or erasers... this may need a bit more elaboration. Thats particularly interesting to explore further as some of the differential genes are expressed by quite significant old changes (like CRECD upon KD of CD47; this is truly amazing but also mysterious).
Response 1: Work from several labs has identified signaling functions of CD47 on the cell surface that control nitric oxide/cGMP signaling, heterotrimeric G protein function, mitochondrial homeostasis, autophagy, and expression of transcription factors including Myc, Sox2, Oct 4, and Klf4 in various cell types. Similarly, the primary cilium, of which IFT57 is a regulatory component, regulates multiple signaling pathways in cells.
Comment 2: How come that altered expression of IFT57 results in changes "in cell motility, differentiation, and extracellular matrix regulation including fibronectin" (line 283)." What are the likely regulatory mechanisms affected? Are there any hypotheses? How would the IFT57-"response" genes look if another cell line would be used for analysis? And would there be an overlap to other cell- or tumor types? Is expression of FT57 specific to epithelial cancers (carcinoma) ?
Response 2: We cited a review that covers various cell functions that are regulated by the primary cilium, and these functions can be cell type-specific. Although specific functions of IFT57 in each of these pathways have not been studied, changes in its expression are known to regulate primary cilium function. To provide readers guidance for identifying pathways and relevant genes regulated by changing IFT57 expression in the thyroid cancer cells, we added an Ingenuity Pathway Analysis of the RNAseq data in the new Data Supplement S2.
Expression of IFT57 is not specific to epithelial cancers. Apart from rare cancers with documented amplification of the IFT57 locus, including two ovarian epithelial carcinomas, IFT57 is expressed at similar levels in all cancers (new Figure S1b).
Comment 3: When checking CRACD expression (= one of the IFT57 "target" genes) in the TCGA data base, by using cBioPortal browser, it does NOT pop up as anti-correlated with IFT57 or CD47 in any of the major tumor types, as you would possibly expect for a "IFT57-downregulated target". It appears only slightly anticorrelated with IFT57 (Spearman -0,21).
Response 3: We did not select CRACD to study because it has the strongest correlation. We selected it as the most selective for regulation by IFT57 versus CD47.
Comment 4: Furthermore, how is CRACD expression in cancers, especially in thyroid carcinoma, regulated? Is there anything known about the major potential drivers of its expression (or repression)? It is okay that the authors do not speculate wirldly on how CRACD cold be possibly regulated by a gene that isnt a transcriptional regulator, but this leaves the reader somehow unsatisfied. Which alternatives exist by which a structural gene (IFT57) could have an impact on gene expression (not function) of other genes? Maybe this could be more elaborated. (I do not have a solution, unfortunately).
Response 4: We agree this is an important topic, but unfortunately, the specific functions of CRACD remain largely unexplored, and until recently it was known only as KIAA1211, a gene of unknown function.
smaller things:
Comment 5: the "dont eat me" hypothesis concerning CD47 needs to be elaborated in the text (lines 54/55). (Like this one, from Wikipedia:" CD-47 acts as a don't eat me signal to macrophages of the immune system which has made it a potential therapeutic target in some cancers, and more recently, for the treatment of pulmonary fibrosis". )
Response 5: text was added to clarify
Comment 6: The correlations between CD47 and IFT57 shown in Table 1 should be in some logical order: why not sort by Spearman/Pearson correlation - highest first?
Response 6: The table entries were sorted by Spearman coefficient as suggested
Comment 7: Correlation and Survival data for these genes and tumour types should also be confirmed independently, using another database. Ideally, one that doesnt use exactly the same TCGA data.
In addition, it may be interesting to further explore the survival data. The GEPIA2 browser (GEPIA 2 (cancer-pku.cn), although also using TCGA data, provides very convincing survival plots, and can distinguish readily between overall and disease- or event-free survival, different margins (qartiles, manually selected group assignments), etc. This could show if the survival data shown in Fig. 2 are confirmed. When checking this, I found different survival values compared to those shown in the Fig. 2.
Response 7: As suggested, we examined the survival correlations with CD47 and IFT57 mRNA expression in Figure 2 using the GEPIA2 (Gene Expression Profiling Interactive Analysis) web server (PMID: 31114875), which includes tumors from both TCGA and GTEx (new supplemental Figure S2). Consistent with Figure 2, CD47 or IFT57 mRNA above the median were associated with poor survival in lung squamous carcinomas and low grade gliomas. Survival trended longer in lung adenocarcinomas with CD47 or IFT57 mRNA above the median but did not achieve significance. We note that GEPIA2 uses percentile cutoffs (e.g. median expression) versus the divergence from mean expression used as cutoff in cBioPortal. The resulting differences in numbers of high versus low tumors and the inclusion of GTEx samples can account for the quantitative differences in p-values. We have examined both overall and event free survival for multiple tumor types using cBioPortal but generally limited the presented data to overall survival for simplicity.
Comment 8: Please explain why you think you were not able to get stronger knock-down CD47 and IFT57 when using CRISPR/cas9 ?
Response 8: The multiple signaling functions of the intraflagellar transport complex in cells and regulation of pathways critical for cell survival by CD47 signaling presumably make cells sensitive to acute loss of gene function. Thyroid cancer cells were less sensitive to CD47 than some other cancer cell types we have studied, and complete loss of cell surface CD47 expression was confirmed by flow cytometry for the pool used by RNAseq analysis.